# FreeNoise: Tuning-Free Longer Video Diffusion via Noise Rescheduling

**Haonan Qiu**[1], **Menghan Xia**[*2], **Yong Zhang**[2],
**Yingqing He**[2,3], **Xintao Wang**[2], **Ying Shan**[2], **Ziwei Liu**[*1]

[1]Nanyang Technological University
[2]Tencent AI Lab
[3]Hong Kong University of Science and Technology

## Abstract

With the availability of large-scale video datasets and the advances of diffusion models, text-driven video generation has achieved substantial progress. However, existing video generation models are typically trained on a limited number of frames, resulting in the inability to generate high-fidelity long videos during inference. Furthermore, these models only support single-text conditions, whereas real-life scenarios often require multi-text conditions as the video content changes over time. To tackle these challenges, this study explores the potential of extending the text-driven capability to generate longer videos conditioned on multiple texts. **1)** We first analyze the impact of initial noise in video diffusion models. Then building upon the observation of noise, we propose **FreeNoise**, a tuning-free and time-efficient paradigm to enhance the generative capabilities of pretrained video diffusion models while preserving content consistency. Specifically, instead of initializing noises for all frames, we reschedule a sequence of noises for long-range correlation and perform temporal attention over them by window-based fusion. **2)** Additionally, we design a novel motion injection method to support the generation of videos conditioned on multiple text prompts. Extensive experiments validate the superiority of our paradigm in extending the generative capabilities of video diffusion models. It is noteworthy that compared with the previous best-performing method which brought about $255\%$ extra time cost, our method incurs only negligible time cost of approximately $17\%$. Generated video samples are available at our website: http://haonanqiu.com/projects/FreeNoise.html.

## 1 Introduction

Diffusion models bring breakthrough developments in image generation (Rombach et al., 2022), enabling users without any art background to easily create unique and personalized designs, graphics, and illustrations based on specific textual descriptions. Building upon this success, there is a growing interest in extending this concept to video generation (He et al., 2022; Ge et al., 2023; Blattmann et al., 2023; Wang et al., 2023c; Luo et al., 2023). As targeting for modeling higher dimensional data, video diffusion model demands a notably increased requirement in model capacity and data scale. As a result, current video diffusion models are generally trained on a small number of frames. Consequently, during the inference stage, the quality of the generated video tends to decrease as the length of the video increases due to longer videos are not supervised during the training stage. One straightforward approach is to generate video fragments of the same length as the training videos and then stitch them together, eliminating the training-inference gap. However, this method results in disconnected and incoherent fragments. To address this issue, the fragments can be fused during the denoising process and smoothly connected in the final video (Wang et al., 2023a). However, the long-distance fragments often have a large content gap to fuse and thus it struggles to maintain the content consistency in the long video. Although some auto-regressive-based methods (Villegas et al., 2022) get rid of this problem by progressively generating the next frame, content consistency is still hard to guarantee due to the error accumulation.

---

[*]Corresponding Authors

In VideoLDM(Blattmann et al., 2023), the generated frame depends not only on the initial noise for the current frame but also on the initial noises for all frames. This means that resampling the noise of any frame will significantly influence other frames due to the full interaction facilitated by the temporal attention layers. This makes it challenging to introduce new content while maintaining the main subjects and scenes of the original video. To address this challenge, we inspect the temporal modeling mechanism of VideoLDM, where the temporal attention module is order-independent, whereas the temporal convolution module is order-dependent. Our experimental observation indicates that the per-frame noises serve as a foundation for determining the overall appearance, while their temporal order influences the content built upon that foundation. Motivated by this, we propose FreeNoise, a tuning-free and time-efficient paradigm to achieve longer video inference. The key idea is to construct a sequence of noise frames with long-range correlation and perform temporal attention over them by the way of window-based fusion. It mainly contains two key designs: *Local Noise Shuffling* and *Window Based Attention Fusion*. By applying the local noise shuffling to a sequence of fixed random noise frames for length extension, we achieve a sequence of noise frames with both internal randomness and long-range correlation. Meanwhile, the window-based attention fusion enables the pre-trained temporal attention modules to process frames of any longer length. Particularly, the overlapped window slicing and merging operation only happens in temporal attention while introducing no computation overhead to other modules of the VideoLDM, which benefits the computational efficiency significantly.

In addition, most video generation models (Blattmann et al., 2023; Luo et al., 2023; Ge et al., 2023) only utilize a single-text condition to control the video even when multi-text conditions are given. For instance, the sentence "A man sleeps on the desk and then reads the book" which contains two stages but only one condition will be reflected in the generated video. This limitation arises from the fact that the training dataset usually contains only a single-text condition. However, in a single-shot scene, the main subject usually involves multiple actions. To address the challenge of generating videos based on multiple prompts without tuning the pretrained models, we propose *Motion Injection*. This approach leverages the characteristics of diffusion models, where different time steps recover varying levels of information (image layout, shapes of the objects, and fine visual details) during the denoising process (Patashnik et al., 2023; Zhang et al., 2023). It gradually injects new motion during the time steps associated with object shapes, following the completion of the previous motion. Importantly, this design does not introduce any additional inference time.

Our contributions are summarized as follows: **1)** We investigate the temporal modeling mechanism of video diffusion models and identify the influence of initial noises. **2)** We design a tuning-free paradigm for longer video generation, which outperforms existing state-of-the-art notably in both video quality and computational efficiency. **3)** We propose an effective motion injection approach that achieves multi-prompt long video generation with decent visual coherence.

## 2 RELATED WORK

### 2.1 VIDEO DIFFUSION MODELS

**Latent Diffusion Models (LDM).** Diffusion models (Sohl-Dickstein et al., 2015; Ho et al., 2020) are generative models that formulate a fixed forward diffusion process to gradually add noise to the data $x_0 \sim p(x_0)$ and learn a denoising model to reverse this process. The forward process contains $T$ timesteps, which gradually add noise to the data sample $x_0$ to yield $x_t$ through a parameterization trick:

$$q(x_t|x_{t-1}) = \mathcal{N}(x_t; \sqrt{1-\beta_t}x_{t-1}, \beta_t I), \qquad q(x_t|x_0) = \mathcal{N}(x_t; \sqrt{\bar{\alpha}_t}x_0, (1-\bar{\alpha}_t)I) \quad (1)$$

where $\beta_t$ is a predefined variance schedule, $t$ is the timestep, $\bar{\alpha}_t = \prod_{i=1}^t \alpha_i$, and $\alpha_t = 1 - \beta_t$. The reverse denoising process obtains less noisy data $x_{t-1}$ from the noisy input $x_t$ at each timestep:

$$p_\theta(x_{t-1} \mid x_t) = \mathcal{N}(x_{t-1}; \mu_\theta(x_t, t), \Sigma_\theta(x_t, t)). \quad (2)$$

Here $\mu_\theta$ and $\Sigma_\theta$ are determined through a noise prediction network $\epsilon_\theta(x_t, t)$, which is supervised by the following objective function, where $\epsilon$ is sampled ground truth noise and $\theta$ is the learnable network parameters.

$$\min_\theta \mathbb{E}_{t,x_0,\epsilon} \|\epsilon - \epsilon_\theta(x_t, t)\|_2^2, \quad (3)$$

Once the model is trained, we can synthesize a data $x_0$ from random noise $x_T$ by sampling $x_t$ iteratively. Recently, to ease the modeling complexity of high dimensional data like images, Latent Diffusion Model (LDM) (Rombach et al., 2022) is proposed to formulate the diffusion and denoising process in a learned low-dimensional latent space. It is realized through perceptual compression with an autoencoder, where an encoder $\mathcal{E}$ maps $x_0 \in \mathbb{R}^{3 \times H \times W}$ to its latent code $z_0 \in \mathbb{R}^{4 \times H' \times W'}$ and a decoder $\mathcal{D}$ reconstructs the image $x_0$ from the $z_0$. Then, the diffusion model $\theta$ operates on the image latent variables to predict the noise $\hat{\epsilon}$.

$$z_0 = \mathcal{E}\left(x_0\right), \quad \hat{x_0} = \mathcal{D}\left(z_0\right) \approx x_0, \quad \hat{\epsilon} = \epsilon_\theta(z_t, y, t), \qquad (4)$$

The network is a sequence of the following layers, where $h$ represents the hidden feature in a certain layer and $y$ denotes conditions like text prompts. Conv and ST are residual convolutional block and spatial transformer, respectively.

$$h' = \mathrm{ST}(\mathrm{Conv}(h, t), y), \quad \mathrm{ST} = \mathrm{Proj}_{\mathrm{in}} \circ (\mathrm{Attn}_{\mathrm{self}} \circ \mathrm{Attn}_{\mathrm{cross}} \circ \mathrm{MLP}) \circ \mathrm{Proj}_{\mathrm{out}}, \qquad (5)$$

**Video Latent Diffusion Model (VideoLDM)** (Blattmann et al., 2023) extends LDM to video generation and trains a video diffusion model in video latent space. The $z_0 \in \mathbb{R}^{4 \times N \times H' \times W'}$ becomes 4 dimensions, and $\theta$ consequently becomes temporal-aware architecture consisting of basic layers as the following equation, where Tconv denotes temporal convolutional block and TT denotes temporal transformers, serving as cross-frame operation modules.

$$h' = \mathrm{TT}(\mathrm{ST}(\mathrm{Tconv}(\mathrm{Conv}(h, t)), y)), \quad \mathrm{TT} = \mathrm{Proj}_{\mathrm{in}} \circ (\mathrm{Attn}_{\mathrm{temp}} \circ \mathrm{Attn}_{\mathrm{temp}} \circ \mathrm{MLP}) \circ \mathrm{Proj}_{\mathrm{out}}. \qquad (6)$$

Following the same architecture, some similar text-to-video models have been proposed (Blattmann et al., 2023; Wang et al., 2023b), primarily differing in training strategies or auxiliary designs (such as fps conditioning, image-video joint training, etc.). AlignYourLatent (Blattmann et al., 2023) is designed to train only the temporal blocks based on a pre-trained text-to-image model (i.e., Stable Diffusion (SD) (Rombach et al., 2022)). In contrast, ModelScope (Wang et al., 2023b) is proposed for fully training the entire model with a SD checkpoint pre-loaded.

## 2.2 LONG VIDEO GENERATION.

Generating long videos poses challenges due to the increased complexity introduced by the temporal dimension, resource limitations, and the need to maintain content consistency. Many GAN-based methods (Skorokhodov et al., 2022; Brooks et al., 2022; Ge et al., 2022) and diffusion-based methods (Harvey et al., 2022; Voleti et al., 2022; Yu et al., 2023; He et al., 2022; Yin et al., 2023; Ho et al., 2022) are proposed to generate long videos. Despite their advantages, those approaches necessitate extensive training on large long video datasets. Recently, a tuning-free method, Gen-L-Video (Wang et al., 2023a) is proposed and successfully extends the video smoothly by merging some overlapping sub-segments into a smoothly changing long segment during the denoising process. However, their content consistency lacks preservation due to the large content gap among those sub-segments. Benefiting from the design of noise rescheduling, our paradigm FreeNoise preserves content consistency well in the generated long videos. Meanwhile, Gen-L-Video costs around 255% extra inference time while FreeNoise only costs 17% additional inference time approximately. Another demand in long video generation is multi-prompt control, as a single-text condition is often insufficient to describe content that evolves over time. While some recent works (Yin et al., 2023; He et al., 2023; Wang et al., 2023a) have explored this direction, they introduce a new lens when a new prompt is provided. Phenaki (Villegas et al., 2022) utilizes an auto-regressive structure to generate one-shot long videos under multi-text conditions but suffers from noticeable content variation. In our paradigm, we can generate multiple motions while preserving the main subjects and scenarios.

## 3 METHODOLOGY

Given a VideoLDM pre-trained on videos with a fixed number of $N_{train}$ frames, our goal is to generate longer videos (e.g., $M$ frames where $M > N_{train}$) without compromising quality by utilizing it for inference. We require the generated $M$ video frames to be semantically accurate and temporally coherent. In the following sections, we will first study the temporal modeling mechanism that challenges VideoLDM in generating longer videos. Subsequently, we will introduce our efficient, tuning-free approach to overcome these challenges. To further accommodate multi-prompt settings, we propose a motion injection paradigm to ensure visual consistency.

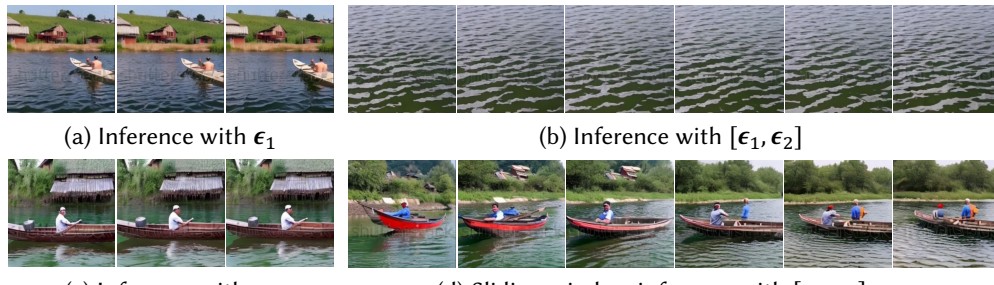

(a) Inference with $\epsilon_1$        (b) Inference with $[\epsilon_1, \epsilon_2]$

(c) Inference with $\epsilon_2$       (d) Sliding window inference with $[\epsilon_1, \epsilon_2]$

Figure 1: Challenges of longer video inference. The random noises $\epsilon_1$ and $\epsilon_2$ have the same number of frames as the model was trained on. All the results are generated under the same text prompt: "*a man is boating on a lake*".

## 3.1 OBSERVATION AND ANALYSIS

**Attentive-Scope Sensitivity.** For longer video generation via VideoLDM, a straightforward solution is to feed $M$ frames of random noises to the model for video generation through iterative denoising steps. Unfortunately, it fails to generate desired result, as the example illustrated in Figure 1(b). The reason is easy to understand: the temporal attention modules perform global cross-frame operations that make all frames attentive to each other, however they are strictly trained to attend on $N_{train}$ neighbor frames and struggle to handle more frames properly. In this case, the generated videos tend to cause semantic incompleteness or temporal jittering.

**Noise-Induced Temporal Drift.** To bypass the issue above, one may argue to employ temporal sliding windows so that the temporal attention module can always process a fixed number of frames. Indeed, this solution makes desired content with a smooth temporal transition. However, it struggles to maintain the long-range visual consistency, as exampled in Figure 1(d). To identify the underlying causes, we explore the temporal modeling mechanism that consists of two kinds of cross-frame operations: temporal attention and temporal convolution. Temporal attention is order-independent, whereas temporal convolution is order-dependent. When temporal convolutions are removed, the output video frames hold a strict correspondence with the initial noise frames, irrespective of shuffling. In contrast, depending on the noise frame order, the temporal convolution introduces new content to ensure the output video's temporal continuity. Figure 2 demonstrates such a phenomena. It implies the conjecture that the per-frame noises serve as a foundation for determining the overall appearance, while their temporal order influences the content built upon that foundation. So, it is challenging for the temporal modules to achieve global coherence when independently sampled noises are combined for longer video generation.

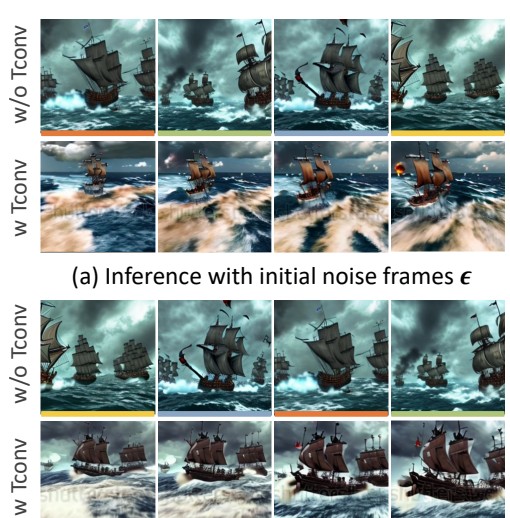

(a) Inference with initial noise frames $\epsilon$

(b) Inference with shuffle($\epsilon$)

Figure 2: Case study on temporal modeling. For the 'w/o Tconv' results in (a) and (b), the frames corresponding to the same initial noise are marked with bottom lines of the same color.

## 3.2 NOISE RESCHEDULING FOR LONG-RANGE CORRELATION

To circumvent the challenges mentioned above, we propose a noise rescheduling paradigm for longer video inference. The key idea is to construct a sequence of noise frames with long-range correlation and perform temporal attention over them by the way of window based fusion. To gain semantically meaningful and visually smooth videos, the model inference should satisfy two basic

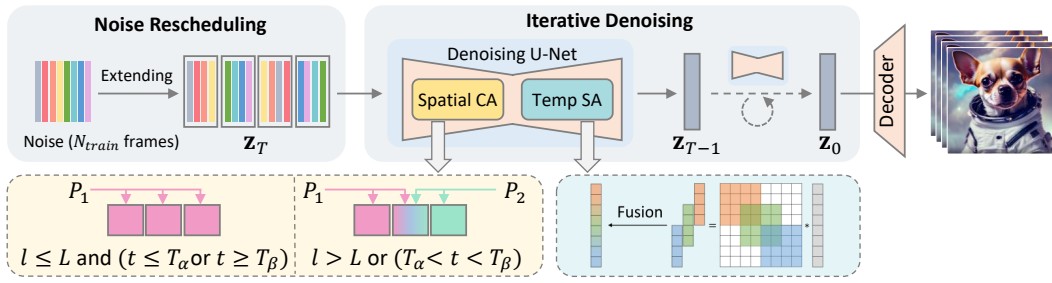

Figure 3: Overview of our proposed method. Given $N_{train}$ frame of random noise, we first extend it to the target $M$ frames as the initial noise $\mathbf{z}_T$ through noise rescheduling. Then, in the iterative denoising process, the multi-prompt injection paradigm is conducted in the spatial cross-attention layers (where $t$ denotes timestep, $l$ denotes layer number, $P$ denotes text prompt) and the sliding window based attention fusion is performed in temporal self-attention layers.

requirements: (i) the temporal attention only accepts fixed $N_{train}$ frames, to bypass the *attentive-scope sensitivity* issue; (ii) every $N_{train}$ frames of features fed to the temporal attention always correspond to $N_{train}$ frames of independent and identically distributed noises, otherwise the generation fails because of the out-of-distribution input. Specifically, we propose two effective designs to achieve this goal.

**Local Noise Shuffle Unit.** To acquire a video with $M$ frames ($M > N_{train}$), we initialize $N_{train}$ frames of random noise $[\epsilon_1, \epsilon_2, ..., \epsilon_{N_{train}}]$ independently and reschedule them for the remaining length:

$$[\epsilon_1, \epsilon_2, ..., \epsilon_{N_{train}}, \text{shuffle}(\epsilon_1, \epsilon_2, ..., \epsilon_S), ..., \text{shuffle}(\epsilon_{\bar{S}_{(i+1)}}, \epsilon_{\bar{S}_{(i+2)}}, ..., \epsilon_{\bar{S}_{(i+S)}}), ...], \quad (7)$$

where $S$ denotes the size of the local shuffle unit and is a divisor of $N_{train}$. $\bar{S}_i = i \bmod N_{train}$, and $i$ is the frame index. The operator $\text{shuffle}(\cdot)$ denotes shuffling the order of the frame sequence. Through such a rescheduling strategy, we achieve a sequence of noise frames with both internal randomness and long-range correlation. Note that, the randomness introduced by temporal shuffle has considerable capacity to bring about content variation, as evidenced by Figure 2.

**Window based Attention Fusion.** Given longer initial noise frames, the spatial modules of VideoLDM process them frame-wisely and the temporal convolution processes the frames in a sliding window, which is the same case as they were trained. Differently, the temporal attention is performed in a global manner and frames longer than $N_{train}$ triggers *attentive-scope sensitivity*. So, we need to deal with the computation of temporal attention so that it can process the longer sequence in the same way as it was trained. Specifically, instead of calculating the temporal attention over all frames, we only calculate temporal attention within each local sliding window of size $U = N_{train}$:

$$F_{i:i+U}^j = \text{Attn}_{\text{temp}}(Q_{i:i+U}, K_{i:i+U}, V_{i:i+U}) = \text{Softmax}\left(\frac{Q_{i:i+U}K_{i:i+U}^T}{\sqrt{d}}\right)V_{i:i+U}, \quad (8)$$

where $i$ is the frame index and $j$ is the window index. Here, we take the sliding stride as the same value as $S$ (the size of noise shuffle unit), so that each sliding window just covers $N_{train}$ frames of independent and identically distributed noises, i.e. $\{\epsilon_1, \epsilon_2, ..., \epsilon_{N_{train}}\}$ with a shuffled order. Figure 3 illustrates the diagram of our attention computation. As each frame is involved in the attention computation of multiple local windows, we need to fuse these attentive outputs to achieve a smooth temporal transition. According to our experiments, naively taking the average will cause dramatic variation in the boundaries of windows. Therefore, we propose to fuse the window-based outputs in a temporal smooth manner, namely computing the weighted sum by taking the frame index distance from each window center as weights:

$$F_i^o = \sum_j \frac{F_i^j * (\frac{U}{2} - \lfloor |i - c^j| \rfloor)}{\sum_j (\frac{U}{2} - \lfloor |i - c^j| \rfloor)}, \quad (9)$$

where $|\cdot|$ denotes absolute value, and $c^j$ is the central frame index of the $j$-th window that covers frame $i$. $F^o$ is the output of the current temporal attention layer. Note that, the overlapped window

slicing and merging operation only happen in temporal attention while introducing no computation overhead to other modules of the U-Net, which benefits the computational efficiency significantly.

### 3.3 Motion Injection for Multi-Prompt Video Generation

Since the aforementioned inference paradigm enables the generation of longer videos, it is natural to explore the potential for synthesizing videos with continuously changing events by utilizing multiple text prompts. This is more challenging because the generation process introduces additional varying factors (i.e. text prompts) that affect the video content mostly. In LDMs, changing a text prompt with only one verb can lead to totally different video content, even with the same initial noises used (Cao et al., 2023). Regarding this, we propose a *motion injection strategy* to modulate the influence of multiple text prompts on video generation content. The key idea is to generate the whole video with the first prompt at most denoising steps (more correlated to scene layout and appearances) and use the target prompt only at some specific steps (more correlated to object shapes and poses).

In VideoLDM, text prompts are taken through the cross-attention mechanism:

$$\widetilde{F} = \text{Attn}_{\text{cross}}(\widetilde{Q}, \widetilde{K}, \widetilde{V}), \widetilde{Q} = l_{\widetilde{Q}}\left(\widetilde{F}_{\text{pre}}\right), \widetilde{K} = l_{\widetilde{K}}(P), \widetilde{V} = l_{\widetilde{V}}(P), \tag{10}$$

where $\widetilde{F}_{\text{pre}}$ is the intermediate features of the network, $P$ is the text embedding by CLIP (Radford et al., 2021a) encoder, and $l_{\widetilde{Q}}, l_{\widetilde{K}}, l_{\widetilde{V}}$ are learned linear layers. According to recent research works (Balaji et al., 2022; Cao et al., 2023), LDMs synthesize different levels of visual content—scene layout, shapes of the objects, and fine details, in the early, middle, and late steps of the denoising process respectively. In our scenarios, we expect the overall layout and object appearance to be similar across prompts while the object poses or shapes should follow the target text prompts. To this end, we gradually inject new motion through the cross attention layer during the time steps associated with object shapes, denoted as $[T_\alpha, T_\beta]$. For the sake of simplicity, we present our method in the case of two text prompts:

$$\textbf{Motion Injection} := \begin{cases} \text{Attn}_{\text{cross}}\left(\widetilde{Q}, l_{\widetilde{K}}(\widetilde{P}), l_{\widetilde{V}}(\widetilde{P})\right), & \text{if } T_\alpha < t < T_\beta \text{ or } l > L, \\ \text{Attn}_{\text{cross}}(\widetilde{Q}, l_{\widetilde{K}}(P_1), l_{\widetilde{V}}(P_1)), & \text{otherwise} \end{cases} \tag{11}$$

$$\widetilde{P} = \begin{cases} P_1, & \text{if } n < N_\gamma, \\ P_1 + \frac{n - N_\gamma}{N_\tau - N_\gamma}(P_2 - P_1), & \text{if } N_\gamma \le n < N_\tau, \\ P_2, & \text{otherwise} \end{cases} \tag{12}$$

where $P_i$ denotes the $i$-th prompt; $\widetilde{P}$ denotes the target prompt of motion injection, which depends on the frame index $n$, and the frames between $[N_\gamma, N_\tau]$ will be assigned with the linearly interpolated embedding to achieve smooth transition; $l > L$ denotes the last $L$ cross-attention layers of the U-Net (e.g. the decoder part). It means that the decoder part will always be provided with the target prompt $\widetilde{P}$ across all the denoising steps, because the decoder features are more tightly aligned with the semantic structures as observed in MasaCtrl (Cao et al., 2023).

## 4 Experiments

**Setting up.** We conduct experiments based on an open-source T2V diffusion model VideoCrafter (Chen et al., 2023) for both singe-prompt and multi-prompt longer video generations. The video diffusion model is trained on 16 frames and is required to sample 64 frames in the inference stage. The window and stride size are set to $U = 16, S = 4$ as default.

**Evaluation Metrics**. To evaluate our paradigm, we report Frechet Video Distance (FVD) (Unterthiner et al., 2018), Kernel Video Distance (KVD) (Unterthiner et al., 2019) and Clip Similarity (CLIP-SIM) (Radford et al., 2021b). Since the longer inference methods are supposed to keep the quality of the original fixed-length inference, we calculate the FVD between original generated short videos and subset generated longer videos with corresponding lengths. CLIP-SIM is used to measure the content consistency of generated videos by calculating the semantic similarity among adjacent frames of generated videos.

Table 1: Quantitative comparison on longer video generation.

| Method | FVD ($\downarrow$) | KVD ($\downarrow$) | CLIP-SIM ($\uparrow$) | Inference Time ($\downarrow$) |
|---|---|---|---|---|
| Direct | 737.61 | 359.11 | 0.9104 | **21.97s** |
| Sliding | 224.55 | 44.09 | 0.9438 | 36.76s |
| GenL | 177.63 | 21.06 | 0.9370 | 77.89s |
| Ours | **85.83** | **7.06** | **0.9732** | 25.75s |

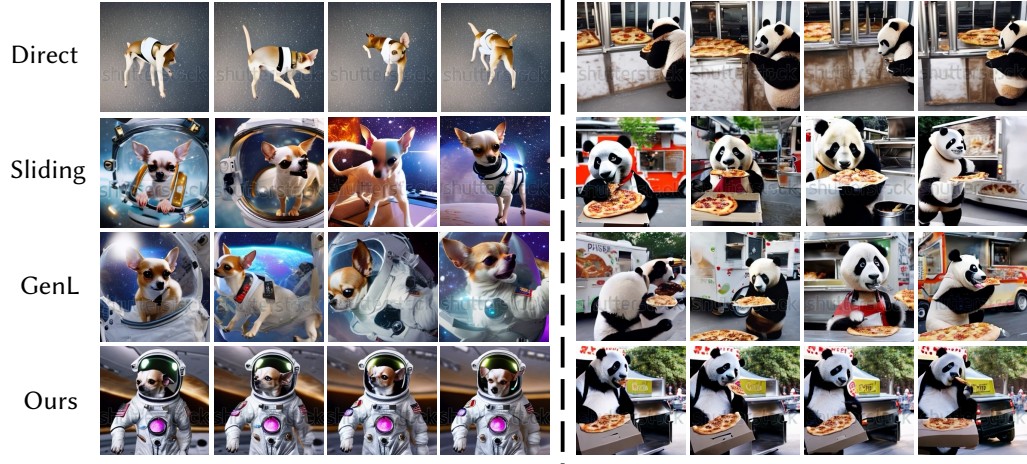

Figure 4: Qualitative comparisons of longer video generation. Left prompt: "*A chihuahua in astronaut suit floating in space, cinematic lighting, glow effect*". Right prompt: "*A very happy fuzzy panda dressed as a chef eating pizza in the New York street food truck*".

## 4.1 LONGER VIDEO GENERATION

We mainly compare our proposed FreeNoise to other tuning-free longer video generation methods with diffusion models. We first directly sample 64 frames (**Direct**). Then we adopt temporal sliding windows so that the temporal attention module can always process a fixed number of frames (**Sliding**). The closest work to our paradigm is Gen-L-Video (**GenL**), which extends the video smoothly by merging some overlapping sub-segments during the denoising process.

The synthesis results are shown in Figure 4. In the first line, the dog has severe artifacts and the background of space is not clear. Obviously, directly sampling 64 frames through a model trained on 16 frames will bring poor quality results due to the training-inference gap. When we use temporal sliding windows, the training-inference gap is eliminated thus more vivid videos are generated. However, this operation ignores the long-range visual consistency thus the resulting subject and background both look significantly different among different frames. Gen-L-Video promotes the integration of frames by averaging the overlapping sub-segments and performs better in some cases. However, it fails to maintain long-range visual consistency and suffers from content mutation. Benefiting from noise rescheduling, all sub-segments in our paradigm share similar main subjects and scenarios while still containing considerable content variation, keeping our main content even when the generated video becomes longer. Results shown in Figure 4 exhibit that our FreeNoise successfully renders high-fidelity longer videos, outperforming all other methods.

In addition, we also compare the operation time of those methods on NVIDIA A100. As presented in Table 1, it is observed that Gen-L-Video exhibits the longest inference time, nearly four times longer than direct inference. This is primarily attributed to its default setting, which involves the nearly global sampling of the entire set of latents four times. However, our paradigm only brings less than 20% extra inference time by limiting most additional calculations within the temporal attention layers.

Table 1 shows quantitative results. The quality of videos generated by direct inference is extremely damaged by the training-inference gap, obtaining the worst FVD and KVD. The video quality from the sliding method and Gen-L-Video is obviously improved but still worse than the results generated by FreeNoise. Our FreeNoise also gains the best CLIP-SIM, indicating the superiority of our method in content consistency.

Table 2: User study. Users are required to pick the best one among our proposed FreeNoise with the other baseline methods in terms of content consistency, video quality, and video-text alignment.

| Method | Content Consistency | Video Quality | Video-Text Alignment |
|---|---|---|---|
| Direct | 11.73% | 10.80% | 11.11% |
| Sliding | 6.17% | 6.79% | 8.02% |
| GenL | 24.38% | 26.85% | 29.63% |
| Ours | **57.72%** | **55.56%** | **51.23%** |

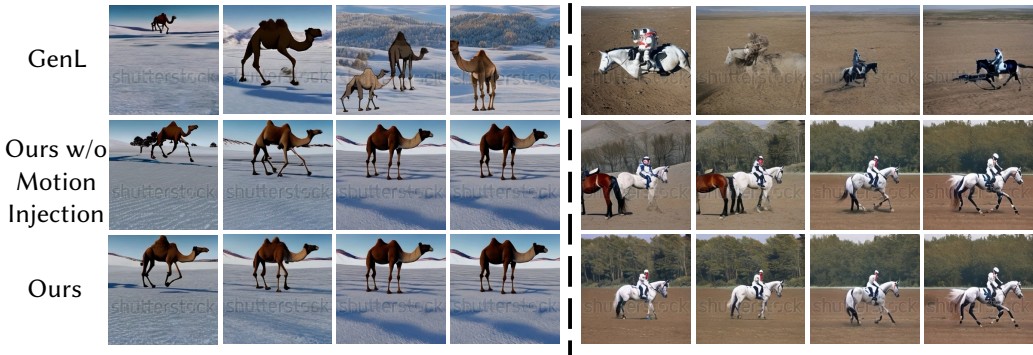

Figure 5: Qualitative comparisons of multi-prompt video generation. Left multi-prompt: "*A camel running on the snow field*" → "*A camel standing on the snow field*". Right multi-prompt: "*An astronaut resting on a horse*" → "*An astronaut riding a horse*".

In addition, we conducted a user study to evaluate our results by human subjective perception. Users are asked to watch the generated videos of all the methods, where each example is displayed in a random order to avoid bias, and then pick up the best one in three evaluation aspects. As shown in Table 2, our approach achieves the highest scores for all aspects: content consistency, video quality, and video-text alignment, outperforming baseline methods by a large margin. Especially for content consistency, our method has received almost twice as many votes as the second place.

**Multi-prompt Video Generation.** We extend our paradigm for multi-prompt video generation by introducing the Motion Injection method. As shown in Figure 5, our method achieves coherent visual coherence and motion continuity: The camel gradually changes from running to standing while the distant mountains remain consistent appearances; The astronaut changes from resting on a horse to riding a horse naturally. However, when we purely use the strategy of noise rescheduling without motion injection, the scene will undergo unexpected changes because a new prompt often introduces unexpected new contents other than the text description due to the inherent properties of the Stable Diffusion model. But it can still work in some cases when the main objects and scenarios are not obviously changed by the new prompt (like the bigfoot case in Figure 5). We also compare with the existing tuning-free state-of-the-art method Gen-L-Video. Figure 5 shows that Gen-L-Video also achieves the conversion of two actions. However, due to the drawbacks of content mutation, its generated objects and scenarios are meaninglessly changed over time.

## 4.2 ABLATION STUDY

**Ablation for Noise Rescheduling.** As noise rescheduling plays an essential role in our method, we typically conduct an ablation to validate its importance, namely removing it from our proposed inference paradigm. In addition, we also implement another variant of our method with the local noise shuffle unit size as $S = 8$ (the sliding window stride is also changed to 8 accordingly).

As shown in Figure 6, without noise rescheduling, our method fails to keep content consistent. Although each frame still matches the text description, they are not semantically connected. And when the sliding window stride is 8, the synthesized features across windows are interacted in a less tight manner. For example, the shape of the bowl is changed gradually in Figure 6. Since the stride value of 4 is able to achieve enough content consistency, we do not consider the smaller stride value of 2, which will bring the double extra inference time.

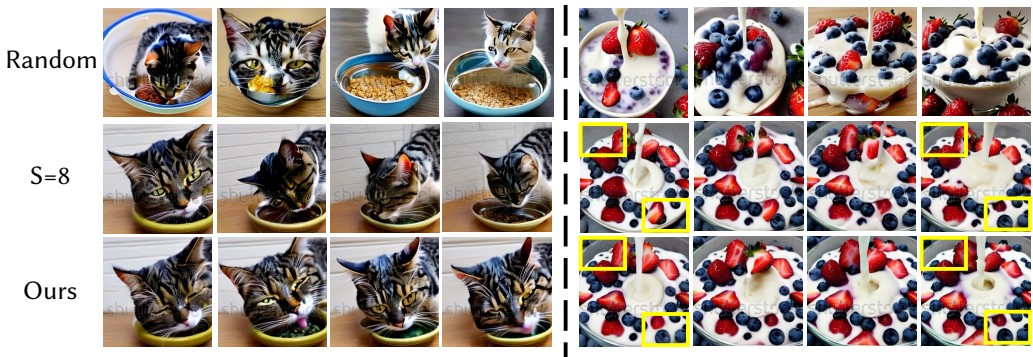

Figure 6: Ablation study on noise rescheduling. Left prompt: "*A cat eating food out of a bowl*". Right prompt: "*A video of milk pouring over strawberries, blueberries, and blackberries*".

**Ablation for Motion Injection.** To show the effectiveness of our design choices in Motion Injection, we study on two main hyper-parameters—layer selection and timestep selection, as expressed in Equation 11. In our design, the last $L$ cross-attention layers of the U-Net (e.g. the decoder part) will always be provided with the target prompt $\widetilde{P}$ across all the denoising steps and $P_1$ only maintains the layout and visual details through the cross-attention layers before the $L$ in some denoising steps. For comparison, we construct another two variations that $P_1$ control the only decoder part and all layers, respectively. Besides, the third variation allows $P_1$ to control the only decoder part across all the denoising steps. Figure 7 shows the results of those variations. When $P_1$ is only allowed to control the decoder part, the running motion of the horse is suppressed. Meanwhile, the appearance of the horse is changed obviously (shown in Figure 7(a)). When $P_1$ is only allowed to control both the encoder and the decoder part, the appearance of the horse is kept but the running mo-

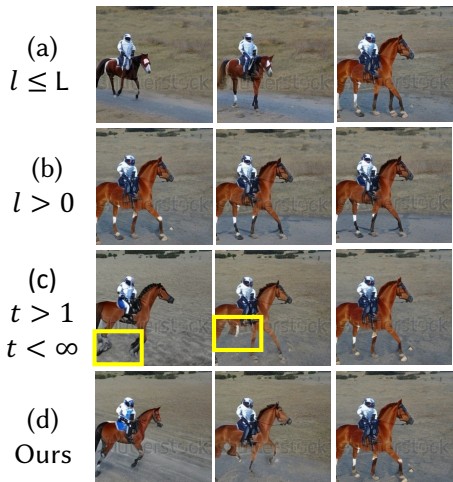

Figure 7: Ablation study on motion injection. The multi-prompt is: "*An astronaut riding a horse*" → "*An astronaut resting on a horse*".

tion is still suppressed (shown in Figure 7(b)). It is because the decoder features are more tightly aligned with the semantic structures, as observed in MasaCtrl (Cao et al., 2023). Compared to layer factors, the choice of timesteps has relatively less influence. Even if we allow $P_1$ to control the only decoder part across all the denoising steps, the horse can still switch smoothly from running to standing. However, this behavior influences the generation of layout and visual details. As a result, a section of the horse's leg will suddenly disappear (shown in Figure 7(c)), while this missing leg appears bent and curled up in the same frame of our final result (shown in Figure 7(d)). Therefore, the selection of time steps can help to achieve more precise control over the content level.

## 5 CONCLUSION

In this study, we addressed the limitations of current video generation models trained on a limited number of frames and supporting only single-text conditions. We explored the potential of extending text-driven generative models to generate high-fidelity long videos conditioned on multiple texts. Through analyzing the impact of initial noise in video diffusion models, we proposed a tuning-free and time-efficient paradigm to enhance the generative capabilities of pretrained models while maintaining content consistency. Additionally, we introduced a novel motion injection method to support multi-text conditioned video generation. Extensive experiments confirmed the superiority of our paradigm in extending the generative capabilities of video diffusion models. Notably, our method achieved this while incurring only approximately 17% additional time cost, compared to the previous best-performing method that required a 255% extra time cost.

## 6 ETHICS STATEMENT

The primary objective of this project is to empower individuals without specialized expertise to create video art more effectively. Our paradigm, based on the pretrained video diffusion model, assists the model in generating longer videos. It is important to note that the content generated by our tuning-free paradigm remains rooted in the original model. As a result, regulators only need to oversee the original video generation model to ensure adherence to ethical standards, and our algorithm does not introduce any additional ethical concerns.

## 7 REPRODUCIBILITY STATEMENT

We have introduced the algorithm and implementation details in detail in the paper. A researcher familiar with the video diffusion model should be able to basically reproduce our method. In addition, we have implemented our FreeNoise on three advanced video generation models.

- VideoCrafter (Chen et al., 2023): https://github.com/AILab-CVC/FreeNoise.
- AnimateDiff (Guo et al., 2023): https://github.com/arthur-qiu/FreeNoise-AnimateDiff.
- LaVie (Wang et al., 2023d): https://github.com/arthur-qiu/FreeNoise-LaVie.

## 8 ACKNOWLEDGEMENTS

This research is supported by the National Research Foundation, Singapore under its AI Singapore Programme (AISG Award No: AISG2-PhD-2022-01-035T), the Ministry of Education, Singapore, under its MOE AcRF Tier 2 (MOE-T2EP20221- 0012) and NTU NAP.

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

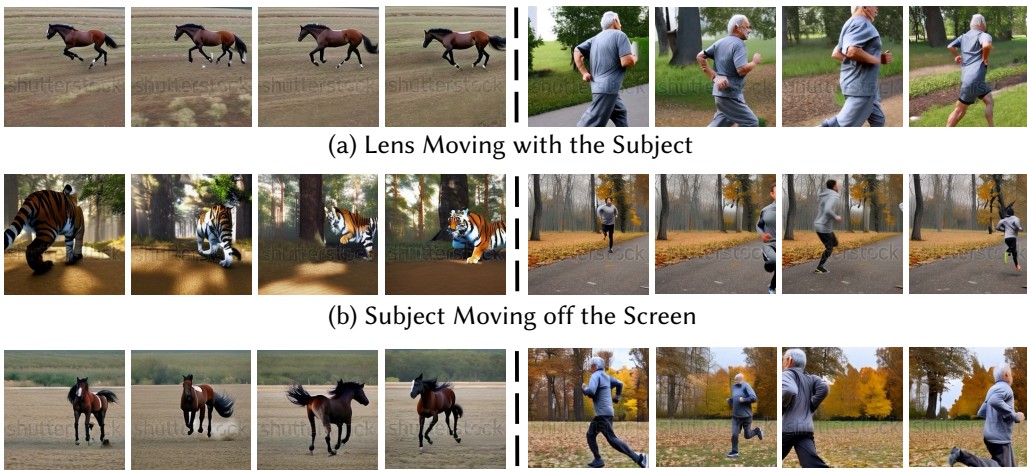

(a) Lens Moving with the Subject

(b) Subject Moving off the Screen

(c) Subject Moving within the Screen

Figure 8: FreeNoise can produce three types of videos exhibiting significant movement: (a) the lens moving with the subject, (b) the subject moving off the screen, and (c) the subject moving within the screen.

# A    APPENDIX: IMPLEMENTATION DETAILS

During sampling, we perform DDIM sampling (Song et al., 2020) with 50 denoising steps, setting DDIM eta to 0. The inference resolution is fixed at $256 \times 256$ pixels. The scale of the classifier-free guidance is set to 15.

For quantitative comparison, we generate a total of 2048 videos for each longer inference method, utilizing 512 prompts (from a standard evaluation paper EvalCrafter (Liu et al., 2023)) and initializing with 4 random initial noises. For calculating the FVD and KVD, we use videos generated by direct inference with training length (16 frames) as the reference set. To align the video length, we cut each long video (64 frames) into 4 segments with 16 frames. Correspondingly, we generate the same number (8192) of short videos with direct inference.

In the user study, we mixed our generated videos with those generated by the other three baselines. A total of 27 users were asked to pick the best one according to the content consistency video quality and video-text alignment, respectively.

# B    APPENDIX: CASE ANALYSIS OF SIGNIFICANT MOVEMENT

Videos with significant movement can be mainly divided into three types:

**Lens Moving with the Subject.** For videos of this type, the position of the subject does not change much and the movement is shown through the regression of the background (Figure 8(a)).

**Subject Moving off the Screen.** For videos of this type, the subject will move off the screen (Figure 8(b)). However, the subject will suddenly appear again due to semantic constraints.

**Subject Moving within the Screen.** For videos of this type, the subject will move within the screen. Due to the size limitation of the screen, the subject will turn around (Figure 8(c)). However, the current pretrained model behaves unnaturally when turning (even for original inference).

As shown in Figure 8, FreeNoise is able to produce all three types. These three types are automatically determined during the inference stage based on the sampled random noises and the given prompt.

## C  APPENDIX: LIMITATION DISCUSSION

As repeated locally shuffled noises are used, FreeNoise has a weakening effect on introducing new content to the video as the length increases. In some cases, the displacement of the subject is limited due to this weakening effect of FreeNoise. However, FreeNoise does not obliterate motion variation or thoroughly fix the spatial structure of objects, such as an object moving from left to right on the screen.

Currently the base model (inference without FreeNoise) only works well with videos of the lens moving with the subject and still struggles to deal with the videos of the subject moving off/within the screen effectively. Therefore, the performance of FreeNoise is also constrained by the base model. We look forward to applying FreeNoise to more powerful video models in the future.

