# OpenReview forum: "FreeNoise: Tuning-Free Longer Video Diffusion via Noise Rescheduling"
_ICLR.cc/2024/Conference — ICLR 2024 poster_

### Official Review · Reviewer_17jd · 2023-10-26

**Soundness:** 3 good
**Presentation:** 3 good
**Contribution:** 2 fair
**Rating:** 6
**Confidence:** 4

**Summary:**

This paper presents a training-free method for extending capabilities of short video diffusion models to generate temporally coherent, longer videos, as well as incorporate multi-text conditioning. The authors propose a novel noise schedule and fused window-based temporal attention to enable more in-distribution, coherent longer generations. In order to enable multi-text conditions, the authors introduce a motion injection method based on conditioning different text prompts at different stages of diffusion sampling.

**Strengths:**

- The paper is well written, and easy to understand
- The proposed noise schedule + temporal attention modification is interesting as a method to enforce better long-term consistency in video
- Incorporating the proposed method is fairly simple, as it does not require any extra training on a pretrained short text-video diffusion model
- Experiments are concrete, and show much quality generations compared baselines

**Weaknesses:**

- From looking at the generated videos, although the proposed method can more cleanly generate longer videos, it seems that the spatial structure of the video (e.g. location of a cat) is very similar throughout the entire video. I believe this may be due to the repetitive nature of shuffled noise reptitions which are generally highly correlated with the structure of the resulting video. So it seems that the method may have a hard time generating more dynamic changes in long videos, such as a cat walking across the screen or scene / camera changes. Could the authors comment on this, or if there are generated video examples with larger structural changes through the video?

**Questions:**

- How would the proposed window fusing (weighted by frame index distance) compared to a simpler scheme such as just merging the current and/or prior window (i.e. similar to a one or two hot version of the frame index weighting).
- Have the authors explored other noise augmentation other than shuffling? In general, shuffling does not seem to be the most intuitive method for perturbing gaussian noise, as it also constrains the epsilons (per frame) to the original samples.
- Section 3.1 mentions that "temporal attention is order independent". Does this imply that the VideoLDM does not have temporal positional embeddings? Or how would it be order independent if it did?

---

> ### Author Response · Authors · 2023-11-18
> **Response to Reviewer 17jd**
>
> Thanks for your valuable comments. We summarize and answer your questions below.
>
> > **Q1: Spatial structure of the video (e.g. location of a cat) is very similar throughout the entire video**
>
> In some cases, the displacement of the subject is limited due to the reused noises of FreeNoise. However, FreeNoise does not obliterate motion variation or thoroughly fix the spatial structure of objects. Please refer to the common response for more details.
>
> > **Q2: Advantages of the proposed weighted window fusion compared to the naive version that uses one-hot weights**
>
> The naive version tends to cause abrupt content changes at window boundaries, and the weighted window fusion can suppress such artifacts by smoothing the temporal transition.
>
> > **Q3: Exploration of other noise augmentation other than shuffling**
>
> We have explored some other strategies in our early experiments. We have tried mixed noise and progressive noise to make the fragments generated by each window more correlated [1]. However, it brings poor quality results due to the training-inference gap. In addition, we have tried to flip noise frames spatially. Although it brings more new content, abrupt changes in content are also introduced. We have uploaded video results to the anonymous website: https://free-noise.github.io/ (A: Other Noise Scheduling).
>
> > **Q4: Clarification on the statement that temporal attention is order independent**
>
> Sorry for the confusion. Through the paper, we use a pre-trained video diffusion model, whose temporal blocks consist of temporal convolution and temporal Transformer with no temporal positional embeddings used. However, if the temporal Transformer uses temporal positional embeddings, the statement that temporal attention is order independent is no longer correct because the position embedding is order sensitive. At that time, temporal attention could be viewed as the superposition of order-independent parts and order-dependent parts. As a result, our method also works for T2V models that use temporal positional embeddings, such as the higher-resolution results (576x1024 model variant of VideoCrafter) demonstrated on our anonymous webpage.
>
> [1] Preserve Your Own Correlation: A Noise Prior for Video Diffusion Models

---

> > ### Comment · Reviewer_17jd · 2023-11-21
> > **Response**
> >
> > Thank you for your response. From the further samples provided it seems that although the model can generate dynamic motion, it can only really do so within the training horizon of the base video prediction model, and the method has more trouble generating consistent dynamic motion across longer temporal resolutions (e.g. locations of entities "snapping back" to their starting locations). However, it is still able to better generate slight variations of motion and background across time compared to baseline methods, and could be useful for potential future applications. My concerns were only partially addressed, so I will maintain my score (6).

---

> ### Author Response · Authors · 2023-11-21
> **Explanation of "snapping back"**
>
> Thanks for your impartial and reasonable evaluation. While our FreeNoise is effective in most aspects, it cannot fully address some of the inherent defects in the base T2V model. The "snapping back" phenomenon is also present in the base model (without FreeNoise) if the subject moves off the screen because the text prompt restricts that there must be an on-screen subject existing (e.x. the description that "a running horse" does not hold if the horse disappears). In real videos, it is common that the lens moves with the subject. When the subject moves off the screen, usually a new clip with a new scene will appear. Consequently, the T2V model lacks data supervision to learn how to let the subject come back naturally. In demonstrated examples shown by recent state-of-the-art EMU VIDEO[1], most movements are also described as "the lens moving with the subject".
>
> In the current stage, a reasonable solution to the "snapping back" phenomenon is generating a new with a new scene after the subject moves off the screen. It can be achieved by either (1) Instant switching: manually switching to a new video clip or (2) gradient transition: sampling new noise and using the new prompt for the frames after the subject moves off the screen.
>
> We hope this explanation provides clarity on the observed phenomenon. We sincerely appreciate your insightful suggestions and recognition of our paper. We also look forward to seeing more advanced video models developed to apply FreeNoise in the future.
>
>
> [1] EMU VIDEO: Factorizing Text-to-Video Generation by Explicit Image Conditioning

---

### Official Review · Reviewer_C7UC · 2023-10-29

**Soundness:** 3 good
**Presentation:** 2 fair
**Contribution:** 2 fair
**Rating:** 5
**Confidence:** 4

**Summary:**

This paper aims to extend the generative capabilities of pre-trained video diffusion models without incurring significant computational costs. It contains three different components, window-based temporal attention, noise rescheduling, motion injection.

**Strengths:**

1. The proposed method is cleverly represented and very easy to follow.
2. The proposed method is much more efficient than the baseline method Gen-L [1].
3. The observation and Analysis part is well-designed and inspiring, and I appreciate this section.



[1] Gen-L-Video: Multi-Text to Long Video Generation via Temporal Co-Denoising

**Weaknesses:**

1. Demo quality. From the demos, we can see that the motions of most generated results are restricted. For example in the ablation study, when the horse running, the background and the position of the horse do not change (even though the legs are moved), which is not a reasonable motion. Therefore, I would say that the proposed method corrupts the motions of the original diffusion models.

2. Many ideas of the paper are used in previous works already. (1) For noise rescheduling, Reuse and Diffuse [1] proposed to reuse previous noise in the later frames. (2) For window-based temporal attention, Align Your Latents [2] applies a similar idea for long video generation, which does not change the temp conv part, but uses sliding local attention to resue the trained temporal attention. I think there's no intrinsic difference. (3) The motion injection part: interpolating the context is already proposed in Gen-L [3].

3. The author says they picked only 100 prompts for quantitative experiments and then generated 2400 videos. This statement seems not explicit.


[1] Reuse and Diffuse: Iterative Denoising for Text-to-Video Generation
[2] Align your Latents: High-Resolution Video Synthesis with Latent Diffusion Models
[3] Gen-L-Video: Multi-Text to Long Video Generation via Temporal Co-Denoising

**Questions:**

See Weaknesses.

---

> ### Author Response · Authors · 2023-11-18
> **Response to Reviewer C7UC**
>
> Thanks for your valuable comments. We summarize and answer your questions below.
>
> > **Q1: Demo quality: motions of most generated results are restricted**
>
> In some cases, the movement of the subject is limited due to the reused noises of FreeNoise. However, FreeNoise does not obliterate motion variation or thoroughly fix the spatial structure of objects. Please refer to the common response for more details.
>
> > **Q2: Many ideas of the paper are used in previous works already.**
>
> (1) Reusing previous noise in later frames (Reuse and Diffuse [1]): [1] extends the noises by simply adding new noises to the reused noises which is significantly different with our noise rescheduling. In practice, the noise reusing scheme of [1] is infeasible to our proposed method because newly added noises will hurt content consistency across frames. Besides, [1] can not be regarded as a previous work since its first submission to arXiv was on **Sept 7th** and the submission date of ICLR is **Sept 28th**.
>
> (2) Sliding local attention (Align Your Latent [2]): this is a commonly used strategy, which can not address the longer video generation itself, as illustrated in Figure 1 (d). In Table 1 and Figure 4, we treat it as one baseline and our results are significantly better than it. In FreeNoise, we divide the whole clips into several overlapped windows with a fixed stride. When the stride = 1, it degenerates into naive sliding attention. However, stride = 1 means that we can not conduct noise rescheduling, causing repetitive results. Therefore, we need to use window-based attention (stride > 1), which is then combined with our proposed weighted fusion of local window attention and noise rescheduling, making FreeNoise effective and non-trivial.
>
> (3) Context interpolation (Gen-L [3]): Simply applying textual interpolation is infeasible to achieve multi-prompt video generation. Our contribution lies in the novel insight and thorough exploration of how to make it work, i.e. using a basic prompt through certain layers and timesteps, then injecting the target prompt for the rest.
>
> > **Q3: The experiment statement is unclear: 2400 videos are generated from 100 prompts for quantitative evaluation**
>
> Sorry for the confusion. Each prompt is sampled 6 times with different random noises. For calculating the FVD and KVD between longer generated videos (64 frames) and normal inference (16 frames), we cut the long video into 4 segments with 16 frames. Therefore, a total of 2400 videos are used for evaluation for each method.
> We have clarified this statement in the revision. In addition, following the suggestion of Reviewer pTfs, we now use 512 prompts from a standard evaluation paper EvalCrafter [4] to rerun the evaluation. Each prompt is sampled four times with different initial noises and a total of 2048 videos are generated for each inference method.
>
> [1] Reuse and Diffuse: Iterative Denoising for Text-to-Video Generation
> [2] Align your Latents: High-Resolution Video Synthesis with Latent Diffusion Models
> [3] Gen-L-Video: Multi-Text to Long Video Generation via Temporal Co-Denoising
> [4] EvalCrafter: Benchmarking and Evaluating Large Video Generation Models

---

> ### Author Response · Authors · 2023-11-21
> **Follow-up Reply**
>
> We sincerely appreciate your great efforts in reviewing this paper. Your constructive advice and valuable comments really help improve our paper. Considering the approaching deadline, please, let us know if you have follow-up concerns. We sincerely hope you can consider our reply in your assessment, and we can further address unclear explanations and remaining concerns if any.
>
> Once more, we appreciate the time and effort you've dedicated to our paper.

---

> ### Comment · Reviewer_C7UC · 2023-11-22
>
> Thank you for your feedback.
> Here is my concern,
>
> 1. About sliding local attention in align your latents, the videos presented in [align-your-latents](https://research.nvidia.com/labs/toronto-ai/VideoLDM/) has much better stability and visual quality in adjacent frames than the sliding baseline proposed in this paper. So I think the author wrongly implement the sliding baseline and not achieves the fair comparison result.
>
> 2.  The authors changed their base model for several times. Previously, they said they use the baseline of VideoLDM. Then, they changed the base model in paper into VideoCrafter. While the experimental results remain the same, making the experimental results not reliable.
>
> 3. The author said simple interpolation of text embedding is infeasible to achieve multi-text conditioned generation, which seems to deliberately belittles the results of Gen-L. We could see from its project page that it do achieve smooth contexts transition in some videos.
>
> Therefore, I think the author didn't strictly and fairly implement the baseline comparisons, making the whole experimental results not reliable.
>
> I decide to degrade my rate to reject temporally.

---

> > ### Author Response · Authors · 2023-11-22
> > **Response to New Concerns of Reviewer C7UC**
> >
> > We are glad to solve your previous concerns and thank you for your in-time response thus we can address your new concerns.
> >
> > > **1. Wrongly implementation of the sliding baseline.**
> >
> > First of all, most videos in Align Your Latent does not use sliding window. Most of their results are 113 frames (Section 4.2 of Align Your Latent). They are generated by one keyframe model with two interpolation processes by the additional model (113 = 7 * 4 * 4 + 1, details in Table 6 of Align Your Latent).
> >
> > The sliding window is described in Appendix D of the paper (paragraph 3). It said that "we find that convolutional-in-time generation can be fragile, in particular when targeting long videos", which also indicates this simple strategy can not tackle long video generation effectively. Only two related video results are shown on their webpage (in the part of Convolutional-in-Time Synthesis). Some degradation in quality can be observed even though those two examples only extend the video less than double. In our paper, the length generated by the sliding window was four times the original, larger degradations should be reasonable.
> >
> > > **2. Changing the base model several times.**
> >
> > We **never** change the base model across all the experiments. In our first version at submission, there is no technical report provided for the open-sourced codebase of VideoCrafter, so we cite a published work (i.e. VideoLDM) that uses the same model architecture to provide a reference. Now that the report of VideoCrafter has been released, we changed the citation for precise clarification. To avoid confusion, we have only updated the pdf **once** during the response period.
> >
> > We apologize for any misunderstanding caused. To demonstrate the reliability of our results, we provided the code (download checkpoint from VideoCrafter) to the Supplementary Material, and the reviewer can check our implementation and test the performance.
> >
> > > **3. About the results of Gen-L.**
> >
> > Gen-L adopts textual embedding interpolation directly, which suffers from notable content changes across frames. On their project page, they showcased six examples that illustrated this weakness exactly, i.e. either background changes or foreground object changes. This is why they prefer to showcase with appearance-changed prompts, which can bypass this weakness to some extent. However, our method is designed to achieve content and object consistency across frames even with multi-prompts, which outperforms Gen-L significantly.
> >
> > In addition, two cases of "jeep car" and two cases of "skiing/surfing" obviously uses additional condition (e.x. depth map) as input which is not aligned with our purely text2video setting. One of the remaining case "A man is boating, village ➜ A man is walking by, city, sunset" use the same text2video setting and abrupt content changes can be observed. The same phenomenon is also observed in the first line of Figure 5 in our main paper.

---

### Official Review · Reviewer_pTfs · 2023-10-30

**Soundness:** 3 good
**Presentation:** 3 good
**Contribution:** 2 fair
**Rating:** 6
**Confidence:** 4

**Summary:**

This paper proposes a method to unlock the long video generation ability of a pretrained text-to-video generation model. The major technical components of this method include: 1. The analysis of artifacts and causes when generating long videos. 2. A noise schedule for long video generation. 3. Windowed attention fusion to keep attention perception field while avoiding content jump between windows, 4. Motion injection for varied textual prompts. The experiments show FreeNoise is a competitive long video generator.

**Strengths:**

This is a technically solid paper. Long video generation is a tricky long-standing problem. The authors propose a series of insights and techniques that have sufficient novelty to address the difficulties:
1.	The analysis of long video artifacts and causes is valuable for developing better video generators.
2.	The noise scheduling and window-based attention fusion address the long video difficulties mentioned in the analysis. They are simple yet effective. Window-based attention fusion addresses the notorious content jump problem, which will likely help develop future video generation foundation models.
3.	FreeNoise does not require additional UNet forward propagation. Therefore, the inference cost overshoot is low.
4.	The qualitative results are marvelous. In human preference evaluation, FreeNoise still achieves the best. The authors provide an anonymous website to show more visual results. The image definition and motion consistency of FreeNoise are both good.
5.	The motion injection technique successfully preserves video contents and drives the video to follow a new text prompt.
6.	The qualitative ablations show each technical component of FreeNoise is effective and important.

**Weaknesses:**

Major concerns:
1.	I’m interested in detailed experiment settings. Please include diffusion sampler configurations, sample resolutions, frame stride, etc. in your future revision.
2.	Please add a pipeline figure. It is not very easy to fully understand how and where FreeNoise is working on generating long videos.
3.	Is direct inference, sliding, GenL, and FreeNoise sharing the same pretrained text-to-video model? If it is, then the evaluation is very convincing since they can all generate the same short video using the prompts but only FreeNoise can achieve good long video results.

Minor concerns:
1.	Page 7. In the second line. A full stop is missing before ‘Obviously’.
2.	In which case FreeNoise may fail? A discussion over the limitations is welcomed.
3.	100 evaluation prompts have limited diversity. If it is feasible, please add more evaluation prompts to make the comparison more convincing.

**Questions:**

1.	The authors claim FreeNoise achieves the maximum long video of 512 frames due to GPU memory limit. Is it possible to unlock even long video generation ability by using the CPU offload technique?
2.	With the help of ControlNet, is it possible to generate more diverse motion with FreeNoise?

---

> ### Author Response · Authors · 2023-11-18
> **Response to Reviewer pTfs**
>
> Thanks for your valuable comments. We summarize and answer your questions below.
>
> > **Q1: Detailed experiment settings**
>
> Thanks for your advice. We have added more experiment details in our revised paper. Due to space limitations, we have moved this part to the appendix.
>
> > **Q2: Are direct inference, sliding, GenL, and FreeNoise sharing the same pretrained text-to-video model?**
>
> Yes. All the methods are evaluated with the same pre-trained T2V model. In addition, their inference parameters (e.x. DDIM steps, scale of classifier-free guidance) are all consistent except for the design of the method itself.
>
> > **Q3: In which case FreeNoise may fail? A discussion over the limitations is welcomed.**
>
> FreeNoise has certain limitations in introducing new content along with the frame length increasing because repeated noises are used. Besides, FreeNoise is a tuning-free approach, it inherits the video generation quality from the base T2V model while can not address the originally existing failure cases.
>
> > **Q4: More evaluation prompts for comparison**
>
> Thanks for your advice. We use 512 prompts from a standard evaluation paper EvalCrafter [1] to rerun the evaluation. Each prompt is sampled four times with different initial noises and totally 2048 videos are generated for each inference method. The updated results are provided in the revised paper (Table 1).
>
> > **Q5: Unlock even longer video generation by using CPU offload technique**
>
> It is feasible but the inference time is crazy for the CPU. In the future, we will explore some technologies to reduce computation [2] and that will be slightly friendly to the CPU.
>
> > **Q6: With the help of ControlNet, is it possible to generate more diverse motion with FreeNoise?**
>
> This is an interesting idea. We have added a depth2video demo at the anonymous website: https://free-noise.github.io/ (B6: Depth2Video). FreeNoise works for ControlNet and additional condition helps to generate more diverse motions. However, naively applying FreeNoise with ControlNet can not work perfectly, because the frame-wise variated depth conditions introduce extra variation to the context. It requires further exploration to make this combination work properly.
>
> [1] EvalCrafter: Benchmarking and Evaluating Large Video Generation Models
> [2] LCM-LoRA: A Universal Stable-Diffusion Acceleration Module

---

> ### Author Response · Authors · 2023-11-21
> **Follow-up Reply**
>
> We sincerely appreciate your great efforts in reviewing this paper. Your constructive advice and valuable comments really help improve our paper. Considering the approaching deadline, please, let us know if you have follow-up concerns. We sincerely hope you can consider our reply in your assessment, and we can further address unclear explanations and remaining concerns if any.
>
> Once more, we appreciate the time and effort you've dedicated to our paper.

---

### Official Review · Reviewer_q7Nz · 2023-10-31

**Soundness:** 3 good
**Presentation:** 3 good
**Contribution:** 3 good
**Rating:** 6
**Confidence:** 4

**Summary:**

This paper investigates the effect of initial noise on a diffusion model for video generation, and thus proposes a method to extend the ability of a pre-trained model to generate long videos without fine-tuning, by rescheduling the initial noise of the video frame and by using a window-based temporal attention to achieve long-range visual consistency. finally, a new method of injecting motion trajectories is proposed, which allows the model to generate videos in response to multiple text prompt.

**Strengths:**

1) The proposed method is simple and effective to expand the model's ability to generate long videos without fine-tuning the model.

2) The use of noise reschedule and window-based attention fusion allows for more consistent video generation.

3) Motion inject allows the model to be fed with a variety of text prompts to generate longer videos with richer meanings.

**Weaknesses:**

1) The method described in this paper lacks suitable diagrams to help illustrate it.

2) The proposed NOISE RESCHEDULING may limite the content variances of video generation since longer videos are produced by repeating the noises for the short ones. As is shown by the examples, the generated long videos looks like a short video that loops multiple times. I wonder whether this way can produce authentic long videos that contain continous various motions.

**Questions:**

Try adding more diagrams to better explain the methods in the article, such as noise rescheduling and an overview of the pipeline for generating a video using the methods mentioned in the article.

---

> ### Author Response · Authors · 2023-11-18
> **Response to Reviewer q7Nz**
>
> Thanks for your valuable comments. For your raised concerns, please refer to our responses to the "Common Response" and "Ability to Generate Videos with Significant Movement". And we have added the pipeline diagram to help understand FreeNoise in the first revision (Figure 3).

---

> > ### Author Response · Authors · 2023-11-21
> > **Follow-up Reply**
> >
> > We sincerely appreciate your great efforts in reviewing this paper. Your constructive advice and valuable comments really help improve our paper. Considering the approaching deadline, please, let us know if you have follow-up concerns. We sincerely hope you can consider our reply in your assessment, and we can further address unclear explanations and remaining concerns if any.
> >
> > Once more, we appreciate the time and effort you've dedicated to our paper.

---

### Author Response · Authors · 2023-11-18
**Ability to Generate Videos with Significant Movement**

As repeated locally shuffled noises are used, FreeNoise has a weakening effect on introducing new content to the video as the length increases. We argue that altering the noise order still allows the generation of certain new content, as observed in Figure 2 (b) (also B1 at the anonymous website: https://free-noise.github.io/).

In some cases, the displacement of the subject is limited due to this weakening effect of FreeNoise. However, FreeNoise does not obliterate motion variation or thoroughly fix the spatial structure of objects, such as an object moving from left to right on the screen. For instance, our proposed method can produce three types of videos exhibiting significant movement: (1) the lens moving with the subject; (2) the subject moving off the screen; and (3) the subject moving within the screen. These three types are automatically determined during the inference stage based on the sampled random noises and the given prompt. Since the base model (inference without FreeNoise) struggles to deal with the other two types effectively, we have only showcased examples of the lens moving with the subject in our previous results. Please see the discussion in the revised Appendix B and video results at the anonymous website: https://free-noise.github.io/ (B: Case Analysis of Significant Movement).

---

### Author Response · Authors · 2023-11-18
**Common Response**

We sincerely thank all reviewers for their constructive suggestions and recognition of our work. We are encouraged that reviewers find our proposed method "simple and effective" (Reviewer q7Nz, pTfs and 17jd); our observation and analysis are "well-designed and inspiring" (Reviewer C7UC); our results "show many quality generations compared baselines" (Reviewer 17jd). We have prepared a common response to clarify the ability to generate videos with significant movement and separate responses for each reviewer.

We have also updated our submission to include the following changes according to reviewers' feedback. Note that the main revisions in the main paper and appendix are highlighted in blue:
- We have added the pipeline diagram to help understand FreeNoise in the first revision (Figure 3).
- Advised by Reviewer pTfs, we now evaluate 512 prompts for quantitative results. Each prompt is sampled four times with different initial noises and a total of 2048 videos are generated for each inference method.
- According to the advice from reviewers, we have added three parts to the Appendix: Implementation Details (Section A), Case Analysis of Significant Movement (Section B), and Limitation Discussion (Section C).
- Some small errors are fixed.

Please do not hesitate to let us know if you have any additional comments or there are more clarifications that we can offer.

---

### Meta-Review · Area_Chair_sfxF · 2023-12-06

**Metareview:**

The paper receives borderline reviews. Most reviewers appreciate the simple yet effective design for tuning-free video generation, and interesting multi-prompt long video generation capabilities. However, one reviewer also raises some doubts regarding the comparison to baselines. In particular, the number of frames generated by the implementation of Align Your Latents is different. Since there is no official implementation available, and the quality is not significantly lower, AC believes this is not a major issue. Reviewers also raised concerns about the ability to generate videos with large motions. The authors provided some examples, though also acknowledged this might be a limitation. The authors are encouraged to address it in the future work.

**Justification For Why Not Higher Score:**

The output quality is not significantly better than previous works, and the individual components are not particularly novel.

**Justification For Why Not Lower Score:**

The paper tries to solve a challenging task, namely multi-prompt long video generation, which could be interesting to the community.

---

### Decision · Program_Chairs · 2024-01-16

Accept (poster)